# Polygenic hazard score is associated with prostate cancer in multi-ethnic populations

Genetic models for cancer have been evaluated using almost exclusively European data, which could exacerbate health disparities. A polygenic hazard score (PHS$_1$) is associated with age at prostate cancer diagnosis and improves screening accuracy in Europeans. Here, we evaluate performance of PHS$_2$ (PHS$_1$, adapted for OncoArray) in a multi-ethnic dataset of 80,491 men (49,916 cases, 30,575 controls). PHS$_2$ is associated with age at diagnosis of any and aggressive (Gleason score $\geq$ 7, stage T3-T4, PSA $\geq$ 10 ng/mL, or nodal/distant metastasis) cancer and prostate-cancer-specific death. Associations with cancer are significant within European (n = 71,856), Asian (n = 2,382), and African (n = 6,253) genetic ancestries (p < 10$^{-180}$). Comparing the 80$^{th}$/20$^{th}$ PHS$_2$ percentiles, hazard ratios for prostate cancer, aggressive cancer, and prostate-cancer-specific death are 5.32, 5.88, and 5.68, respectively. Within European, Asian, and African ancestries, hazard ratios for prostate cancer are: 5.54, 4.49, and 2.54, respectively. PHS$_2$ risk-stratifies men for any, aggressive, and fatal prostate cancer in a multi-ethnic dataset.

Prostate cancer is the second most common cancer diagnosed in men worldwide, causing substantial morbidity and mortality[1]. Prostate cancer screening may reduce morbidity and mortality[2–5], but to avoid overdiagnosis and overtreatment of indolent disease[6–9], it should be targeted and personalized. Prostate cancer age at diagnosis is important for clinical decisions regarding if/when to initiate screening for an individual[10,11]. Survival is another key cancer endpoint recommended for risk models[12].

Genetic risk stratification is promising for identifying individuals with a greater predisposition for developing cancer[13–16], including prostate cancer[17]. Polygenic models use common variants—identified in genome-wide association studies—whose combined effects can assess the overall risk of disease development[18,19]. Recently, a polygenic hazard score (PHS) was developed as a weighted sum of 54 single-nucleotide polymorphisms (SNPs) that models a man's genetic predisposition for developing prostate cancer[13]. Validation testing was done using ProtecT trial data[2] and demonstrated the PHS to be associated with age at prostate cancer diagnosis, including aggressive prostate cancer[13]. However, the development and validation datasets were limited to men of European ancestry. While genetic risk models might be important clinical tools for prognostication and risk stratification, using them may worsen health disparities[20–24] because most models are constructed using European data and may underrepresent genetic variants important in persons of non-European ancestry[20–24]. Indeed, this is particularly concerning in prostate cancer, as race/ethnicity is an important prostate cancer risk factor; diagnostic, treatment, and outcomes disparities continue to exist between different races/ethnicities[25,26].

Here, we assessed PHS performance in a multi-ethnic dataset that includes individuals of European, African, and Asian genetic ancestry. This dataset also includes long-term follow-up information, affording an opportunity to evaluate PHS for association with fatal prostate cancer.

## Results

**Adaption of PHS for OncoArray.** Of the 30 SNPs from $PHS_1$ not directly genotyped on OncoArray, proxy SNPs were identified for 22 (linkage disequilibrium ≥ 0.94). Therefore, $PHS_2$ included 46 SNPs, in total (Supplementary Information). $PHS_2$ association with age at aggressive prostate cancer diagnosis in ProtecT was similar to that previously reported for $PHS_1$ ($z = 21.7$, $p = 3.6 \times 10^{-104}$ for $PHS_1$; $z = 21.4$, $p = 1.3 \times 10^{-101}$ for $PHS_2$). $HR_{98/50}$ was 4.68 [95% CI: 3.62–6.15] for $PHS_2$, compared to 4.61 [3.52–5.99] for $PHS_1$.

**PHS association with any prostate cancer in OncoArray.** $PHS_2$ was associated with age at prostate cancer diagnosis in all three OncoArray-defined genetic ancestry groups (Table 1). Comparing the 80th and 20th percentiles of genetic risk, men with high PHS had an HR of 5.32 [4.99–5.70] for any prostate cancer. Within

each genetic ancestry group, men with high PHS had HRs of 5.54 [5.18–5.93], 4.49 [3.23–6.33], and 2.54 [2.08–3.10] for men of European, Asian, and African ancestry, respectively.

**PHS association with aggressive prostate cancer in OncoArray.** $PHS_2$ was associated with age at aggressive prostate cancer diagnosis in all three OncoArray-defined genetic ancestry groups (Table 2). Comparing the 80th and 20th percentiles of genetic risk, men with high PHS had an HR of 5.88 [5.46–6.33] for aggressive prostate cancer; within each genetic ancestry group, men with high PHS had HRs of 5.62 [5.23–6.05], 5.16 [4.79–5.55], and 2.43 [2.26-2.61] for men of European, Asian, and African ancestry, respectively.

**PHS association with fatal prostate cancer in OncoArray.** $PHS_2$ was associated with age at prostate cancer death for all men in the multi-ethnic dataset ($z = 15.9$, $p = 6.3 \times 10^{-57}$). Table 3 shows z-scores and corresponding HRs for fatal prostate cancer. Comparing the 80th and 20th percentiles of genetic risk, men with high PHS had a HR of 5.68 [5.07–6.46] for prostate cancer death.

**Sensitivity analyses.** Sensitivity analyses demonstrated that large changes in assumed population incidence had minimal effect on the calculated HRs for any, aggressive, or fatal prostate cancer (Supplementary Information).

**PHS and family history.** Family history was also associated with any prostate cancer ($z = 39.7$, $p < 10^{-300}$; Table 4), aggressive prostate cancer ($z = 32.4$, $p = 2.7 \times 10^{-230}$), and fatal prostate cancer ($z = 8.76$, $p = 1.4 \times 10^{-18}$) in the multi-ethnic dataset. Among those with known family history, the combination of family history and PHS performed better than family history alone (log-likelihood $p < 10^{-300}$). This pattern held true when analyses were repeated on each genetic ancestry. Additional family history analyses are reported in the Supplementary Information.

**PHS associations with aggressive prostate cancer using alternative ancestry groupings**

*Agnostic genetic ancestry groupings with fastSTRUCTURE.* With fastSTRUCTURE, the optimal model was the one with $K = 2$ clusters: cluster 1 had mainly men of European OncoArray-defined genetic ancestry and self-reported race/ethnicity, cluster 2 had only men of African OncoArray-defined genetic ancestry and mostly Black/African American self-reported race/ancestry, while the Admixed cluster included men of all Oncotype-defined genetic ancestries. Table 5 demonstrates the $HR_{80/20}$ for aggressive prostate cancer for these $K = 2$ fastSTRUCTURE-defined clusters. Comparing the 80th and 20th percentiles of genetic risk, men with high PHS had HRs for aggressive prostate cancer of 5.60 [5.55, 5.64], 2.06 [2.03, 2.09], and 5.05 [4.89, 5.21] for

### Table 1 Association of PHS with prostate cancer.

| OncoArray genetic ancestry | z (p Value) | Hazard ratios [95% CI] comparing percentiles of $PHS_2$ | | | |
|---|---|---|---|---|---|
| | | $HR_{20/50}$: ≤20th vs. 30–70th | $HR_{80/50}$: ≥80th vs. 30–70th | $HR_{98/50}$: ≥98th vs. 30–70th | $HR_{80/20}$: ≥80th vs. ≤20th |
| All ($n = 80,491$) | 54.3 ($p < 10^{-300}$) | 0.45 [0.43-0.46] | 2.39 [2.31-2.47] | 4.21 [3.99-4.47] | 5.32 [4.99-5.70] |
| European ($n = 71,856$) | 55.8 ($p < 10^{-300}$) | 0.44 [0.43-0.45] | 2.44 [2.35-2.53] | 4.34 [4.09-4.60] | 5.54 [5.18-5.93] |
| Asian ($n = 2382$) | 46.7 ($p < 10^{-300}$) | 0.48 [0.40-0.56] | 2.15 [1.81-2.57] | 3.77 [2.80-5.13] | 4.49 [3.23-6.33] |
| African ($n = 6253$) | 28.7 ($p = 3.8 \times 10^{-181}$) | 0.63 [0.57-0.69] | 1.59 [1.44-1.76] | 2.27 [1.91-2.71] | 2.54 [2.08-3.10] |

Hazard ratios (HRs) are shown comparing men in the highest 2% of genetic risk (≥98th percentile of PHS), highest 20% of genetic risk (≥80th percentile), average risk (30–70th percentile), and lowest 20% of genetic risk (≤20th percentile) across genetic ancestry. p Values reported are two-tailed from the Cox models.

**Table 2 Association of PHS with aggressive prostate cancer.**

| OncoArray genetic ancestry | z (p Value) | Hazard ratios [95% CI] comparing percentiles of PHS$_2$ | | | |
|---|---|---|---|---|---|
| | | HR$_{20/50}$: ≤20th vs. 30–70th | HR$_{80/50}$: ≥80th vs. 30–70th | HR$_{98/50}$: ≥98th vs. 30–70th | HR$_{80/20}$: ≥80th vs. ≤20th |
| All (n = 58,600) | 47.6 (p < 10$^{-300}$) | 0.43 [0.41–0.44] | 2.50 [2.42–2.60] | 4.61 [4.33–4.90] | 5.88 [5.48–6.34] |
| European (n = 53,608) | 46.4 (p < 10$^{-300}$) | 0.44 [0.42–0.45] | 2.45 [2.36–2.55] | 4.40 [4.15–4.70] | 5.62 [5.25–6.05] |
| Asian (n = 1806) | 43.8 (p < 10$^{-300}$) | 0.45 [0.37–0.55] | 2.32 [1.88–2.89] | 4.14 [2.92–6.03] | 5.16 [3.45–7.78] |
| African (n = 3186) | 23.6 (p = 7.2 × 10$^{-123}$) | 0.64 [0.49–0.81] | 1.55 [1.23–2.00] | 2.18 [1.44–3.43] | 2.43 [1.51–4.05] |

Hazard ratios (HRs) derived from Cox proportional hazards models are shown comparing men in the highest 2% of genetic risk (≥98th percentile of PHS), highest 20% of genetic risk (≥80th percentile), average risk (30–70th percentile), and lowest 20% of genetic risk (≤20th percentile) across genetic ancestry. p Values reported are two-tailed from the Cox models.

**Table 3 Association of PHS with death from prostate cancer.**

| Ancestry | z (p Value) | Hazard ratios [95% CI] comparing percentiles of PHS$_2$ | | | |
|---|---|---|---|---|---|
| | | HR$_{20/50}$: ≤20th vs. 30–70th | HR$_{80/50}$: ≥80th vs. 30–70th | HR$_{98/50}$: ≥98th vs. 30–70th | HR$_{80/20}$: ≥80th vs. ≤20th |
| All (n = 78,221) | 15.9 (p = 6.3 × 10$^{-57}$) | 0.43 [0.41–0.56] | 2.47 [2.33–2.64] | 4.46 [4.04–4.98] | 5.68 [5.07–6.46] |

Hazard ratios (HRs) from Cox proportional hazards models are shown comparing men in the highest 2% of genetic risk (≥98th percentile of PHS), highest 20% of genetic risk (≥80th percentile), average risk (30–70th percentile), and lowest 20% of genetic risk (≤20th percentile). p Values reported are two-tailed from the Cox models.

**Table 4 Multivariable models with both PHS and family history of prostate cancer (≥1 first-degree relative affected) for association with any prostate cancer in the multi-ethnic dataset, and by genetic ancestry.**

| OncoArray genetic ancestry | Variable | beta | z-score | p Value | HR |
|---|---|---|---|---|---|
| All (n = 46,030) | PHS | 1.98 | 53.3 | <10$^{-300}$ | 4.48 |
| | Family history | 0.94 | 38.6 | <10$^{-300}$ | 2.55 |
| European (n = 39,445) | PHS | 2.06 | 56.2 | <10$^{-300}$ | 4.80 |
| | Family history | 0.92 | 38.1 | <10$^{-300}$ | 2.50 |
| Asian (n = 1028) | PHS | 1.89 | 50.7 | <10$^{-300}$ | 4.17 |
| | Family history | 0.72 | 21.2 | 9.5 × 10$^{-100}$ | 2.05 |
| African (n = 5557) | PHS | 1.11 | 26.2 | 2.6 × 10$^{-151}$ | 2.22 |
| | Family history | 1.14 | 46.7 | <10$^{-300}$ | 3.11 |

This analysis is limited to individuals with known family history. Both family history and PHS were significantly associated with any prostate cancer in the combined models. Hazard ratios (HRs) for family history were calculated as the exponent of the beta from the multivariable Cox proportional hazards regression[56]. The HR for PHS in the multivariable models was estimated as the HR$_{80/20}$ (men in the highest 20% vs. those in the lowest 20% of genetic risk by PHS$_2$) in each cohort. p Values reported are two-tailed from the Cox models. The model with PHS performed better than family history alone (log-likelihood p < 10$^{-300}$).

**Table 5 Association of PHS with aggressive prostate cancer, by two clusters using fastSTRUCTURE.**

| fastSTRUCTURE K | Cluster | HR$_{80/20}$: ≥80th vs. ≤20th |
|---|---|---|
| K = 2 | 1 | 5.60 [5.55–5.64] |
| | 2 | 2.06 [2.03–2.09] |
| | Admixed | 5.05 [4.89–5.21] |

Hazard ratios (HRs) from Cox proportional hazards models are shown comparing men in the highest 20% of genetic risk (≥80th percentile) vs. the lowest 20% of genetic risk (≤20th percentile).

cluster 1, cluster 2, and admixed cluster, respectively. Corresponding results for the K = 3–6 clustering approaches are shown in the Supplementary Information.

*Self-reported race/ethnicity.* HRs for aggressive prostate cancer comparing the 80th and 20th percentiles of genetic risk when participants are stratified by their self-reported race/ethnicity are shown in the Supplementary Information.

## Discussion

These results confirm the previously reported association of PHS with age at prostate cancer diagnosis in Europeans and show that this finding generalizes to a multi-ethnic dataset, including men of European, Asian, and African ancestry. PHS is also associated with age at aggressive prostate cancer diagnosis and at prostate cancer death. Comparing the highest and lowest quintiles of genetic risk, men with high PHS had HRs of 5.32, 5.88, and 5.68 for any prostate cancer, aggressive prostate cancer, and prostate cancer death, respectively.

We found that PHS is associated with prostate cancer in men of European, Asian, and African genetic ancestry (and a wider range of self-reported race/ethnicities). Current prostate cancer screening guidelines suggest possible initiation at earlier ages for men of African ancestry, given higher incidence rates and worse survival when compared to men of European ancestry[26]. Using the PHS to risk-stratify men might help with decisions regarding when to initiate prostate cancer screening: perhaps a man with African genetic ancestry in the lowest percentiles of genetic risk by PHS could safely delay or forgo screening to decrease the possible harms associated with overdetection and overtreatment[9], while a man in the highest risk percentiles might consider screening at an earlier age. Similar

reasoning applies to men of all genetic ancestries. Risk-stratified screening should be prospectively evaluated.

PHS performance was better in those with OncoArray-defined European and Asian genetic ancestry than in those with African ancestry. For example, comparing the highest and lowest quintiles of genetic risk, men with OncoArray-defined European and Asian genetic ancestry with high PHS had HRs for any prostate cancer of 5.54 and 4.49 times, respectively, while the analogous HR for men of African genetic ancestry was 2.54. This trend was also observed for aggressive prostate cancer. Moreover, the optimal fastSTRUCTURE clustering of our dataset ($K = 2$) yielded one cluster that consisted of almost only men of African ancestry (by both self-report and OncoArray-defined genetic ancestry) and had inferior risk stratification with $PHS_2$ (HR 2.06), compared to the performance observed in the other cluster (nearly all European) and an admixed cluster (HRs 5.60 and 5.05, respectively). Overall, these results suggest PHS can differentiate men of higher and lower risk in each ancestral group, but the range of risk levels may be narrower in those of African ancestry. Possible reasons for relatively diminished performance include increased genetic diversity with less linkage disequilibrium in those of African genetic ancestry[27–29]. Known health disparities may also contribute[25], as the availability—and timing—of PSA results may depend on healthcare access. Alarmingly, there has historically been a poor representation of African populations in clinical or genomic research studies[20,21]. This pattern is reflected in the present study, where most men of African genetic ancestry were missing clinical diagnosis information used to determine disease aggressiveness. That such clinical information is less available for men of African ancestry also leaves open the possibility of systematic differences in the diagnostic workup—and therefore the age of diagnosis—across different ancestry populations. These are critical health disparities that will need to be addressed (and ultimately eliminated) to ensure equitable and accurate genomic prostate cancer stratification for all men. Notwithstanding these caveats, the present PHS is associated with age at prostate cancer diagnosis in men of African ancestry, possibly paving the way for more personalized screening decisions for men of African descent. Promising efforts are also underway to further improve PHS performance in men of African ancestry[30].

The first PHS validation study used data from ProtecT, a large prostate cancer trial[2,13]. ProtecT's screening design yielded biopsy results from both controls and cases with PSA ≥ 3 ng/mL, making it possible to demonstrate improved accuracy and efficiency of prostate cancer screening with PSA testing. Limitations of the ProtecT analysis, though, include few recorded prostate cancer deaths in the available data, and the exclusion of advanced cancer from that trial[2]. The present study includes long-term observation, with both early and advanced disease[18], allowing for evaluation of PHS association with any, aggressive, and fatal prostate cancer; we found PHS to be associated with all outcomes.

Age is critical in clinical decisions of whether men should be offered prostate cancer screening[31–34] and in how to treat men diagnosed with prostate cancer[31,32]. Age may also inform prognosis[32,35]. Age at diagnosis or death is therefore of clinical interest in inferring how likely a man is to develop cancer at an age when he may benefit from treatment. One important advantage of the survival analysis used here is that it permits men without cancer at the time of the last follow-up to be censored while allowing for the possibility of them developing prostate cancer (including aggressive or fatal prostate cancer) later on. prostate cancer death is a hard endpoint with less uncertainty than clinical diagnosis (which may vary with screening practices and delayed medical attention). PHS may help identify men with a high (or low) genetic predisposition to develop lethal prostate cancer and could assist physicians in deciding when to initiate screening.

Current guidelines suggest considering a man's individual cancer risk factors, overall life expectancy, and medical comorbidities when deciding whether to screen[6]. The most prominent clinical risk factors used in practice are family history and race/ ethnicity[6,36,37]. Combined PHS and family history performed better than either alone in this multi-ethnic dataset. This finding is consistent with a prior report that PHS adds considerable information over family history alone. The prior study did not find an association of family history with age at prostate cancer diagnosis, perhaps because the universal screening approach of the ProtecT trial diluted the influence of family history on who is screened in typical practice[13]. In the present study, family history and PHS appear complementary in assessing prostate cancer genetic risk. Moreover, the HRs for PHS suggest clinical relevance similar or greater to predictive tools routinely used for cancer screening (e.g., breast cancer) and for other diseases (e.g., diabetes and cardiovascular disease). HRs reported for those tools are around 1–3 for disease development or other adverse outcome[38–42]; HRs reported here for PHS (for any, aggressive, or fatal prostate cancer) are similar or greater.

Limitations to this work include that the dataset comes from multiple, heterogeneous studies, from various populations with variable screening rates. This allowed for a large, multi-ethnic dataset that includes clinical and survival data, but comes with uncertainties avoided in the ProtecT dataset used for original validation. However, the heterogeneity would likely reduce the PHS performance, not systematically inflate the results. Second, we note that no germline SNP tool, including this PHS, has been shown to discriminate men at risk of aggressive prostate cancer from those at risk of only indolent prostate cancer. Third, while the OncoArray-defined and fastSTRUCTURE genetic ancestry classifications used here may be more accurate than self-reported race/ethnicity alone[43] and allowed for evaluation of admixed genetic ancestry, detailed analysis of local ancestry was not assessed. As noted above, clinical data availability was not uniform across contributing studies and was lower in men of OncoArray-defined African genetic ancestry. Efforts to improve genetic risk prediction should focus on consistent data collection patterns and elimination of data disparities so that models are widely applicable for all men. We also found that while the optimal fastSTRUCTURE model had $K = 2$ clusters for risk stratification men for aggressive prostate cancer, models with more $K$ clusters also produced comparable (or larger ranges) of hazard ratios for risk stratification. The ability of these models with more $K$ clusters to risk-stratify men well (while possibly being less representative of the available data) emphasizes the dire need for more complex and deeper studies evaluating the intersection of genetics, the granularity of ancestry, and prostate cancer risk. In addition, the PHS may not include all SNPs associated with prostate cancer; in fact, over 60 additional SNPs have been reported since the development of the original PHS[18]. Some of these SNPs are ethnicity-specific, including within non-European populations[44–46], and will be included in further model optimization to improve prostate cancer risk stratification. Future work could also evaluate the PHS performance in relation to epidemiological risk factors associated with prostate cancer risk beyond those currently used in clinical practice (i.e., family history and race/ethnicity). Finally, various circumstances and disease-modifying treatments may have influenced post-diagnosis survival to an unknown degree. Despite this possible source of variability in survival among men with fatal prostate cancer, PHS was still associated with age at death, an objective, and meaningful endpoint. Future development and optimization hold promise for improving upon the encouraging risk stratification achieved here in men of different genetic ancestries, particularly African.

In summary, PHS was associated with age at any and aggressive prostate cancer, and at death from prostate cancer in a multi-ethnic

dataset. PHS performance was relatively diminished in men of African genetic ancestry, compared to performance in men of European or Asian genetic ancestry. PHS risk-stratifies men of various genetic ancestries for prostate cancer and should be prospectively studied as a means to individualize screening strategies seeking to reduce prostate cancer morbidity and mortality.

## Methods

**Participants.** We obtained data from the OncoArray project[47] that had undergone quality control steps[18]. This dataset includes 91,480 men with genotype and phenotype data from 64 studies (Supplementary Information). Individuals whose data were used in the prior development or validation of the original PHS model (PHS$_1$) were excluded ($n = 10,989$)[13], leaving 80,491 in the independent dataset used here. Table 6 describes available data. Individuals not meeting the endpoint for each analysis were censored at age of last follow-up.

All contributing studies were approved by the relevant ethics committees; written informed consent was acquired from the study participants[48]. The present analyses used de-identified data from the PRACTICAL consortium.

**Polygenic hazard score.** The original PHS$_1$ was validated for association with age at prostate cancer diagnosis in men of European ancestry using a survival analysis[13]. To ensure the score was not simply identifying men at risk of indolent disease, PHS$_1$ was also validated for association with age at aggressive prostate cancer (defined as an intermediate-risk disease, or above[6]) diagnosis[13]. PHS$_1$ was calculated as the vector product of a patient's genotype ($X_i$) for $n$ selected SNPs and the corresponding parameter estimates ($\beta_i$) from a Cox proportional hazards regression:

$$PHS = \sum_{i}^{n} X_i \beta i \tag{1}$$

The 54 SNPs in PHS$_1$ were selected using PRACTICAL consortium data ($n = 31,747$ men) genotyped with a custom array (iCOGS, Illumina, San Diego, CA)[13].

**Adapting the PHS to OncoArray.** Genotyping for the present study was performed using a commercially available, cancer-specific array (OncoArray, Illumina, San Diego, CA)[18]. Twenty-four of the 54 SNPs in PHS$_1$ were directly genotyped on OncoArray. We identified proxy SNPs for those not directly genotyped and recalculated the SNP weights in the same dataset used for the original development of PHS$_1$[13] (Supplementary Methods).

The performance of the adapted PHS (PHS$_2$), was compared to that of PHS$_1$ in the ProtecT dataset originally used to validate PHS$_1$ ($n = 6411$). PHS$_2$ was calculated for all patients in the ProtecT validation set and was tested as the sole predictive variable in a Cox proportional hazards regression model ($R$ v.3.5.1, "survival" package[49]) for age at aggressive prostate cancer diagnosis, the primary endpoint of that study. The performance was assessed by the metrics reported during the PHS$_1$ development:[13] z-score and hazard ratio (HR$_{98/50}$) for aggressive prostate cancer between men in the highest 2% of genetic risk (≥98th percentile) vs. those with average risk (30–70th percentile). HR 95% confidence intervals (CIs) were determined by bootstrapping 1000 random samples from the ProtecT dataset[50,51] while maintaining the same number of cases and controls. PHS$_2$ percentile thresholds are shown in the Supplementary Information.

**OncoArray-defined genetic ancestry.** Self-reported race/ethnicities[47,52], included European, Black, or African American (includes Black African, Black Caribbean), East Asian, South Asian, Hawaiian, Hispanic American, and Other/Unknown.

Genetic ancestry for each individual from the OncoArray project[47] was provided with the PRACTICAL consortium data. Briefly, genotypes from 2318 ancestry informative markers were mapped into a two-dimensional space representing the first two principal components, which has been shown to yield results very similar to those obtained with the STRUCTURE approach[52]. The distance from the individual's mapping to the three reference clusters (European, African, and Asian) was then used to estimate the individual's genetic ancestry[47,52]. Individuals were classified into one of three OncoArray-defined labels; European: greater than 80% European ancestry, Asian: greater than 40% Asian ancestry, and African: greater than 20% African ancestry. Individuals not meeting any of the aforementioned three labels were classified as "other," but all of the individuals in the present prostate cancer dataset met the criteria for one of the three OncoArray-defined genetic ancestries.

**Any prostate cancer.** We tested PHS$_2$ for association with age at diagnosis of any prostate cancer in the multi-ethnic dataset ($n = 80,491$, Table 6).

PHS$_2$ was calculated for all patients in the multi-ethnic dataset and used as the sole independent variable in Cox proportional hazards regressions for the endpoint of age at prostate cancer diagnosis. Due to the potential for Cox proportional hazards results to be biased by a higher number of cases in our dataset than in the general population, sample-weight corrections were applied to all Cox models using population data from Sweden[13,53] (additional details are in Supplementary Information). Significance was set at $\alpha = 0.01$[13].

These Cox proportional hazards regressions (with PHS$_2$ as the sole independent variable and age at prostate cancer diagnosis as the outcome) were then repeated for subsets of data, stratified by OncoArray-defined genetic ancestry: European, Asian, and African. Percentiles of genetic risk were calculated using data from the 9,728 men in the original (iCOGS) development set who were less than 70 years old and without prostate cancer[13,54]. HRs and 95% CIs for each genetic ancestry group were calculated to make the following comparisons: HR$_{98/50}$, men in the highest 2% of genetic risk vs. those with average risk (30–70th percentile); HR$_{80/50}$, men in the highest 20% vs. those with average risk, HR$_{20/50}$, men in the lowest 20% vs. those with average risk; and HR$_{80/20}$, men in the highest 20% vs. lowest 20%. CIs were determined by bootstrapping 1000 random samples from each genetic ancestry group[50,51] while maintaining the same number of cases and controls. HRs and CIs were calculated for age at prostate cancer diagnosis separately for each genetic ancestry group.

Given that the overall incidence of prostate cancer in different populations varies, we performed a sensitivity analysis of the population case/control numbers, allowing the population incidence to vary from 25 to 400% of that reported in Sweden (chosen as an example population; Supplementary Information).

**Aggressive prostate cancer.** Recognizing that not all prostate cancer is clinically significant, we also tested PHS$_2$ for association with age at aggressive prostate cancer diagnosis in the multi-ethnic dataset. For these analyses, we included cases that had known tumor stage, Gleason score, and PSA at diagnosis ($n = 60,617$ cases, Table 6). Aggressive prostate cancer cases were those that met any of the following criteria[6,13]: Gleason score ≥7, PSA ≥ 10 ng/mL, T3–T4 stage, nodal metastases, or distant metastases. As before, Cox proportional hazards models and sensitivity analysis were used to assess the association.

**Fatal prostate cancer.** Using an even stricter definition of clinical significance, we evaluated the association of PHS$_2$ with age at prostate cancer death in the multi-ethnic dataset. All cases (regardless of staging completeness) and controls were included, and the endpoint was the age at death due to prostate cancer. This analysis was not stratified by genetic ancestry due to low numbers of recorded prostate cancer deaths in the non-European datasets. The cause of death was

## Table 6 Participant characteristics, n = 80,491.

| | OncoArray-defined genetic ancestry | | | |
| --- | --- | --- | --- | --- |
| | All | European | Asian | African |
| *Participants* | | | | |
| Controls | 30,575 | 26,377 | 1185 | 3013 |
| Prostate cancer cases | 49,916 | 45,479 | 1197 | 3240 |
| Aggressive prostate cancer cases[a] | 26,419 | 24,279 | 716 | 1424 |
| Fatal prostate cancer cases | 3983 | 3908 | 57 | 18 |
| *Number of participants with known first-degree family history information* | | | | |
| Family history of prostate cancer available (prostate cancer cases; controls) | 46,030 (28,204; 17,826) | 39,445 (24,921; 14,524) | 1,028 (519; 509) | 5,557 (2,764; 2,793) |
| *Age demographics* | | | | |
| Median age, at diagnosis (IQR) | 65 [60–71] | 66 [60–71] | 68 [62–74] | 62 [56–68] |
| Median age, at last follow up (IQR) | 70 [63–76] | 70 [64–77] | 70 [63–76] | 62 [56–68] |

[a]Aggressive prostate cancer defined as: Gleason scores ≥7, PSA ≥ 10 ng/mL, T3–T4 stage, nodal metastases, or distant metastases.
*IQR* interquartile range.

determined by the investigators of each contributing study using cancer registries and/or medical records (Supplementary Information). At last follow-up, 3983 men had died from prostate cancer, 5806 had died from non-prostate cancer causes, and 70,702 were still alive. The median age at the last follow-up was 70 years (IQR: 63–76). As before, Cox proportional hazards models and sensitivity analysis were used to assess the association.

**PHS and family history.** Prostate cancer family history was also tested for association with any, aggressive, or fatal prostate cancer. Information on family history was standardized across studies included in PRACTICAL consortium data. A family history of prostate cancer was defined as the presence or absence of a first-degree relative with a prostate cancer diagnosis. There were 46,030 men with available prostate cancer family history data.

Cox proportional hazards models were used to assess family history for association with any, aggressive, or fatal prostate cancer. To evaluate the relative importance of each, a multivariable model using both family history and PHS was compared to using family history alone (log-likelihood test; $\alpha = 0.01$). HRs were calculated for each variable.

### Explorations of alternative ancestry groupings

*Agnostic genetic ancestry groupings with FastSTRUCTURE.* The primary analyses, above, used OncoArray-defined genetic ancestries, as prior reports have shown genetic ancestry may be more informative than self-reported race/ethnicities[43]. However, for the purpose of this study, the OncoArray-defined categories may underestimate the impact of the inherent complexity of human genetic ancestry. Therefore, we further explored the impact of an array of alternative genetic ancestry subgroup definitions on $PHS_2$ performance using fastSTRUCTURE[55], which infers global admixture/ancestry via a Bayesian approach. We ran fastSTRUCTURE v1.0 on all individuals in the multi-ethnic dataset using approximately 2300 ancestry informative markers and multiple ($K$) levels of population complexity to agnostically cluster the data into $K = 2$–6 populations. For each iteration of $K$ populations, participants were placed into the cluster for which their maximum admixture proportion was ≥0.8. Those participants without a cluster for which their maximum admixture proportion was ≥0.8 were placed into a separate group termed "admixed." The optimal number of clusters ($K$) for fastSTRUCTURE was chosen as that which maximized the marginal likelihood of the data[55]. $PHS_2$ was evaluated for association with aggressive prostate cancer ($HR_{80/20}$) after stratification by each $K$ population subgroup.

A comparison of fastSTRUCTURE clustering, OncoArray-determined genetic ancestry, and self-reported race/ethnicity was compiled. OncoArray-defined genetic ancestry was mostly concordant with self-reported race/ethnicity. Participants with other/unknown self-reported race/ethnicity were mostly grouped into OncoArray's European genetic ancestry. Additional details are shown in the Supplementary Information.

*Self-reported race/ethnicity.* Finally, we also evaluated PHS performance for association with aggressive prostate cancer using participants' self-reported race/ethnicity.

**Reporting summary.** Further information on research design is available in the Nature Research Reporting Summary linked to this article.

## Data availability

PRACTICAL consortium data are available upon request to the Data Access Committee (http://practical.icr.ac.uk/blog/?page_id=135). Questions and requests for further information may be directed to PRACTICAL@icr.ac.uk. All other data are available within the Article, Supplementary information, or upon request to the authors.

## Code availability

Code used for this work has been made available along with this paper (Supplementary Software 1).

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

## Acknowledgements

Acknowledgments for the PRACTICAL consortium and contributing studies are described in the Supplementary Material. This study was funded in part by a grant from the United States National Institute of Health/National Institute of Biomedical Imaging and Bioengineering (#K08EB026503), United States Department of Defense (#W81XWH-13-1-0391), University of California CRCC C21CR2060, the Research Council of Norway (#223273), K.G. Jebsen Stiftelsen, and South East Norway Health Authority. RM Martin is supported in part by the National Institute for Health Research Bristol Biomedical Research Centre. The CAP trial is funded by Cancer Research UK and the UK Department of Health (C11043/A4286, C18281/A8145, C18281/A11326, C18281/A15064, and C18281/A24432). R.M. Martin was supported by a Cancer Research UK (C18281/A19169) program grant (the Integrative Cancer Epidemiology Programme). The views and opinions expressed by authors in this publication are those of the authors and do not necessarily reflect those of the UK National Institute for Health Research (NIHR) or the Department of Health and Social Care. The content is solely the responsibility of the authors and does not necessarily represent the official views of any of the funding agencies, who had no role in the design and conduct of the study; collection, management, analysis, and interpretation of the data; preparation, review, or approval of the manuscript; and decision to submit the manuscript for publication. Funding for the PRACTICAL consortium member studies is detailed in the Supplementary Information.

## Author contributions

M.-P.H.-L., C.C.F., R.K., W.K.T., I.G.M., O.A.A., A.M.D., and T.M.S. designed the study concept, created the methodology, and analyzed/interpreted the data. M.-P.H.-L., M.E.M., R.A.E., Z.K.-J., K.M., J.S., N.P., J.B., H.G., D.E.N., J.L.D., F.C.H., R.M.M., S.F.N., B.G.N., F.W., C.M.T., G.G.G., A.W., D.A., R.C.T., W.J.B., W.Z., M.S., J.L.S., L.A.M., C.M.L.W., A.S.K., O.C., S.I.B., S.K., K.D.S., C.C., E.M.G., F.M., K.-T.K., J.Y.P., S.A.I., C.M., R.J.M., S.N.T., B.S.R., T.-J.L., S.W., A.V., M.K., K.L.P., C.H., M.R.T., L.M., R.J.L., L.C.-A., H.B., E.M.J., R.K., C.J.L., S.L.N., K.D.R., H.P., A.R., L.F.N., J.H.F., M.G., N.U., F.C., M.G.-D., P.A.T., W.S.B., M.J.R., M.E.P., J.J.H., and T.M.S. acquired the data. M.-P.H.-L. and T.M.S. wrote the original drafts of the paper and Supplementary Information. All authors approved the final version of the paper and Supplementary Information.

## Competing interests

A.M. Dale and T.M. Seibert report a research grant from the US Department of Defense. O.A. Andreassen reports research grants from K.G. Jebsen Stiftelsen, Research Council of Norway, and South East Norway Health Authority. N. Usmani reports grants from Astra Zeneca and Astellas, research collaboration, and financial in-kind support from Best Medical Canada and Concure Oncology. R.M. Martin reports grants from Cancer Research UK, during the conduct of the study. K.D. Sørensen reports grants from Danish Cancer Society, grants from Velux Foundation, during the conduct of the study. T.M. Seibert reports honoraria from Multimodal Imaging Services Corporation for imaging segmentation and honoraria from Varian Medical Systems and WebMD, Inc. for educational content. A.S. Kibel reports advisory board memberships for Sanofi-Aventis, Dendreon, and Profound. R.A. Eeles reports honoraria from GU-ASCO, honoraria/speaker fees from Janssen, honoraria from an invited talk to the University of Chicago, and educational honoraria from Bayer&Ipsen. K.D. Sørensen reports personal fees from AstraZeneca, personal fees from Sanofi, outside the submitted work. N. Usmani reports honoraria from Janssen Canada and Bayer, outside the submitted work. M. Gamulin reports speaker/advisor board/travel fees for BMS, Pfizer, Novartis, Astellas, Sanofi, Janssen, Roche, Sandoz, Amgen, Bayer, PharmaSwiss, MSD, Alvogen. M. Gamuli also reports non-financial report for drugs from BMS, Roche, Janssen. A.M. Dale has additional disclosures outside the present work: founder, equity holder, and advisory board member for CorTechs Labs, Inc.; advisory board member of Human Longevity, Inc.; recipient of nonfinancial research support from General Electric Healthcare. K.D. Sørensen is co-inventor on an issued patent ("Biomarkers for prostate cancer"/# US10106854B2, # AU2013275761B2, # JP6242388B2) licensed to Qiagen, on an issued patent ("A microRNA-based method for early detection of prostate cancer in urine samples"/# US10400288B2, # EP3256602B1, # ES2749651T3) licensed to Qiagen, and on an issued patent ("A microRNA-based method for assessing the prognosis of a prostate cancer patient"/# US10358681B2, # EP3262186B1, # ES2724404T3, #JP6769979B2), licensed to Qiagen. N. Usmani has a patent (US Provisional Patent Application No. 62/688,481: "Theranostic radiophotodynamic therapy nanoparticles") pending, and a patent (US Patent Application No. 15/978,996: "Hand-held device and computer-implemented system and method for assisted steering of a percutaneously inserted needle") pending. The remaining authors declare no competing interests. Additional acknowledgments for the PRACTICAL consortium and contributing studies are described in the Supplemental Material.

## Additional information

Minh-Phuong Huynh-Le [1,2], Chun Chieh Fan [2], Roshan Karunamuni[1,2], Wesley K. Thompson[3,4], Maria Elena Martinez[5], Rosalind A. Eeles [6,7], Zsofia Kote-Jarai[6], Kenneth Muir[8,9], Johanna Schleutker [10,11], Nora Pashayan [12,13,14], Jyotsna Batra [15,16], Henrik Grönberg [17], David E. Neal[18,19,20], Jenny L. Donovan[21], Freddie C. Hamdy[22,23], Richard M. Martin [21,24,25], Sune F. Nielsen[26,27], Børge G. Nordestgaard [28,29], Fredrik Wiklund [30], Catherine M. Tangen[31], Graham G. Giles [32,33,34], Alicja Wolk [35,36], Demetrius Albanes[37], Ruth C. Travis [38], William J. Blot[39,40], Wei Zheng [41], Maureen Sanderson[42], Janet L. Stanford[43,44], Lorelei A. Mucci[45], Catharine M. L. West [46], Adam S. Kibel[47], Olivier Cussenot[48,49], Sonja I. Berndt[50], Stella Koutros[50], Karina Dalsgaard Sørensen [51,52], Cezary Cybulski[53], Eli Marie Grindedal[54], Florence Menegaux[55,56], Kay-Tee Khaw[57], Jong Y. Park [58], Sue A. Ingles[59], Christiane Maier[60], Robert J. Hamilton[61,62], Stephen N. Thibodeau[63], Barry S. Rosenstein[64,65], Yong-Jie Lu[66], Stephen Watya[67], Ana Vega [68,69,70], Manolis Kogevinas[71,72,73], Kathryn L. Penney[74], Chad Huff[75], Manuel R. Teixeira [76,77], Luc Multigner [78], Robin J. Leach[79], Lisa Cannon-Albright [80,81], Hermann Brenner[82,83,84], Esther M. John[85], Radka Kaneva[86], Christopher J. Logothetis[87], Susan L. Neuhausen[88], Kim De Ruyck[89], Hardev Pandha[90], Azad Razack[91], Lisa F. Newcomb[43,92], Jay H. Fowke[93,94], Marija Gamulin [95], Nawaid Usmani[96,97], Frank Claessens[98], Manuela Gago-Dominguez[99,100], Paul A. Townsend[101], William S. Bush [102], Monique J. Roobol [103], Marie-Élise Parent[104,105], Jennifer J. Hu[106], Ian G. Mills[107], Ole A. Andreassen [108], Anders M. Dale[2,109], Tyler M. Seibert [1,2,109,110✉], UKGPCS collaborators, APCB (Australian Prostate Cancer BioResource), NC-LA PCaP Investigators, The IMPACT Study Steering Committee and Collaborators, Canary PASS Investigators, The Profile Study Steering Committee & The PRACTICAL Consortium

[1]Department of Radiation Medicine and Applied Sciences, University of California San Diego, La Jolla, CA, USA. [2]Center for Multimodal Imaging and Genetics, University of California San Diego, La Jolla, CA, USA. [3]Division of Biostatistics and Halicioğlu Data Science Institute, University of California San Diego, La Jolla, CA, USA. [4]Department of Family Medicine and Public Health, University of California San Diego, La Jolla, CA, USA. [5]Moores Cancer Center, Department of Family Medicine and Public Health, University of California San Diego, La Jolla, CA, USA. [6]The Institute of Cancer Research, London, UK. [7]Royal Marsden NHS Foundation Trust, London, UK. [8]Division of Population Health, Health Services Research and Primary Care, University of Manchester, Oxford Road, Manchester, UK. [9]Warwick Medical School, University of Warwick, Coventry, UK. [10]Institute of Biomedicine, Kiinamyllynkatu 10, FI-20014 University of Turku, Turku, Finland. [11]Department of Medical Genetics, Genomics, Laboratory Division, Turku University Hospital, Turku, Finland. [12]University College London, Department of Applied Health Research, London, UK. [13]Centre for Cancer Genetic Epidemiology, Department of Oncology, University of Cambridge, Strangeways Laboratory, Worts Causeway, Cambridge, UK. [14]Department of Applied Health Research, University College London, London, UK. [15]Australian Prostate Cancer Research Centre-Qld, Institute of Health and Biomedical Innovation and School of Biomedical Sciences, Queensland University of Technology, Brisbane, QLD, Australia. [16]Translational Research Institute, Brisbane, QLD, Australia. [17]Department of Medical Epidemiology and Biostatistics, Karolinska Institute, Stockholm, Sweden. [18]Nuffield Department of Surgical Sciences, University of Oxford, John Radcliffe Hospital, Headington, Oxford, UK. [19]Department of Oncology, University of Cambridge, Addenbrooke's Hospital, Cambridge, UK. [20]Cancer Research UK, Cambridge Research Institute, Li Ka Shing Centre, Cambridge, UK. [21]Population Health Sciences, Bristol Medical School, University of Bristol, Bristol, UK. [22]Nuffield Department of Surgical Sciences, University of Oxford, Oxford, UK. [23]Faculty of Medical Science, University of Oxford, John Radcliffe Hospital, Oxford, UK. [24]National Institute for Health Research (NIHR) Biomedical Research Centre, University of Bristol, Bristol, UK. [25]Medical Research Council (MRC) Integrative Epidemiology Unit, University of Bristol, Bristol, UK. [26]Faculty of Health and Medical Sciences, University of Copenhagen, Copenhagen, Denmark. [27]Department of Clinical Biochemistry, Herlev and Gentofte Hospital, Copenhagen University Hospital, Herlev, Copenhagen, Denmark. [28]Faculty of Health and Medical Sciences, University of Copenhagen, Copenhagen, Denmark. [29]Department of Clinical Biochemistry, Herlev and Gentofte Hospital, Copenhagen University Hospital, Herlev, Copenhagen, Denmark. [30]Department of Medical Epidemiology and Biostatistics, Karolinska Institute, Stockholm, Sweden. [31]SWOG Statistical Center, Fred Hutchinson Cancer Research Center, Seattle, WA, USA. [32]Cancer Epidemiology Division, Cancer Council Victoria, Melbourne, VIC, Australia. [33]Centre for Epidemiology and Biostatistics, Melbourne School of Population and Global Health, The University of Melbourne, Parkville, VIC, Australia. [34]Precision Medicine, School of Clinical Sciences at Monash Health, Monash University, Clayton, VIC, Australia. [35]Division of Nutritional Epidemiology, Institute of Environmental Medicine, Karolinska Institutet, Stockholm, Sweden. [36]Department of Surgical Sciences, Uppsala University, Uppsala, Sweden. [37]Division of Cancer Epidemiology and Genetics, National Cancer Institute, NIH, Bethesda, MD, USA. [38]Cancer Epidemiology Unit, Nuffield Department of Population Health, University of Oxford, Oxford, UK. [39]Division of Epidemiology, Department of Medicine, Vanderbilt University Medical Center, Nashville, TN, USA. [40]International Epidemiology Institute, Rockville, MD, USA. [41]Division of Epidemiology, Department of Medicine, Vanderbilt University Medical Center, Nashville, TN, USA. [42]Department of Family and Community Medicine, Meharry Medical College, Nashville, TN, USA. [43]Division of Public Health Sciences, Fred Hutchinson Cancer Research Center, Seattle, WA, USA. [44]Department of Epidemiology, School of Public Health, University of Washington, Seattle, WA, USA. [45]Department of Epidemiology, Harvard T. H. Chan School of Public Health, Boston, MA, USA. [46]Division of Cancer Sciences, University of Manchester, Manchester Academic Health Science Centre, Radiotherapy Related Research, The Christie Hospital NHS Foundation Trust, Manchester, UK. [47]Division of Urologic Surgery, Brigham and Womens Hospital, Boston, MA, USA. [48]Sorbonne Universite, GRC n°5, AP-HP, Tenon Hospital, 4 Rue de la Chine, Paris, France. [49]CeRePP, Tenon Hospital, Paris, France. [50]Division of Cancer Epidemiology and Genetics, National Cancer Institute, NIH, Bethesda, MD, USA. [51]Department of Molecular Medicine, Aarhus University Hospital, Aarhus, Denmark. [52]Department of Clinical Medicine, Aarhus University, Aarhus, Denmark. [53]International Hereditary Cancer Center,

Department of Genetics and Pathology, Pomeranian Medical University, Szczecin, Poland. [54]Department of Medical Genetics, Oslo University Hospital, Oslo, Norway. [55]Cancer & Environment Group, Center for Research in Epidemiology and Population Health (CESP), INSERM, University Paris-Sud, University Paris-Saclay, Villejuif Cédex, France. [56]Paris-Sud University, UMRS 1018, Villejuif Cedex, France. [57]Clinical Gerontology Unit, University of Cambridge, Cambridge, UK. [58]Department of Cancer Epidemiology, Moffitt Cancer Center, Tampa, FL, USA. [59]Department of Preventive Medicine, Keck School of Medicine, University of Southern California/Norris Comprehensive Cancer Center, Los Angeles, CA, USA. [60]Humangenetik Tuebingen, Tuebingen, Germany. [61]Department of Surgical Oncology, Princess Margaret Cancer Centre, Toronto, ON, Canada. [62]Department of Surgery (Urology), University of Toronto, Toronto, ON, Canada. [63]Department of Laboratory Medicine and Pathology, Mayo Clinic, Rochester, MN, USA. [64]Department of Radiation Oncology and Department of Genetics and Genomic Sciences, Icahn School of Medicine at Mount Sinai, One Gustave L. Levy Place, New York, NY, USA. [65]Department of Genetics and Genomic Sciences, Icahn School of Medicine at Mount Sinai, New York, NY, USA. [66]Centre for Molecular Oncology, Barts Cancer Institute, Queen Mary University of London, John Vane Science Centre, Charterhouse Square, London, UK. [67]Uro Care, Kampala, Uganda. [68]Fundación Pública Galega Medicina Xenómica, Santiago De Compostela, Spain. [69]Instituto de Investigación Sanitaria de Santiago de Compostela, Santiago De Compostela, Spain. [70]Centro de Investigación en Red de Enfermedades Raras (CIBERER), Santiago De Compostela, Spain. [71]ISGlobal, Barcelona, Spain. [72]IMIM (Hospital del Mar Medical Research Institute), Barcelona, Spain. [73]Universitat Pompeu Fabra (UPF), Barcelona, Spain. [74]Channing Division of Network Medicine, Department of Medicine, Brigham and Women's Hospital/Harvard Medical School, Boston, MA, USA. [75]The University of Texas M. D. Anderson Cancer Center, Houston, TX, USA. [76]Department of Genetics, Portuguese Oncology Institute of Porto (IPO-Porto), Porto, Portugal. [77]Biomedical Sciences Institute (ICBAS), University of Porto, Porto, Portugal. [78]Univ Rennes, Inserm, EHESP, Irset (Institut de Recherche en Santé, Environnement et Travail)—UMR_S 1085, Rennes, France. [79]Department of Urology, Mays Cancer Center, University of Texas Health Science Center at San Antonio, San Antonio, TX, USA. [80]Division of Epidemiology, Department of Internal Medicine, University of Utah School of Medicine, Salt Lake City, UT, USA. [81]George E. Wahlen Department of Veterans Affairs Medical Center, Salt Lake City, UT, USA. [82]Division of Clinical Epidemiology and Aging Research, German Cancer Research Center (DKFZ), Heidelberg, Germany. [83]German Cancer Consortium (DKTK), German Cancer Research Center (DKFZ), Heidelberg, Germany. [84]Division of Preventive Oncology, German Cancer Research Center (DKFZ) and National Center for Tumor Diseases (NCT), Im Neuenheimer Feld 460, Heidelberg, Germany. [85]Department of Medicine, Division of Oncology, Stanford Cancer Institute, Stanford University School of Medicine, Stanford, CA, USA. [86]Molecular Medicine Center, Department of Medical Chemistry and Biochemistry, Medical University of Sofia, Sofia, Bulgaria. [87]The University of Texas M. D. Anderson Cancer Center, Department of Genitourinary Medical Oncology, Houston, TX, USA. [88]Department of Population Sciences, Beckman Research Institute of the City of Hope, Duarte, CA, USA. [89]Ghent University, Faculty of Medicine and Health Sciences, Basic Medical Sciences, Gent, Belgium. [90]The University of Surrey, Guildford, Surrey, UK. [91]Department of Surgery, Faculty of Medicine, University of Malaya, Kuala Lumpur, Malaysia. [92]Department of Urology, University of Washington, Seattle, WA, USA. [93]Department of Medicine and Urologic Surgery, Vanderbilt University Medical Center, Nashville, TN, USA. [94]Division of Epidemiology, Department of Preventive Medicine, The University of Tennessee Health Science Center, Memphis, TN, USA. [95]Department of Oncology, University Hospital Centre Zagreb, University of Zagreb, School of Medicine, Zagreb, Croatia. [96]Department of Oncology, Cross Cancer Institute, University of Alberta, Edmonton, Alberta, Canada. [97]Division of Radiation Oncology, Cross Cancer Institute, Edmonton, Alberta, Canada. [98]Department of Cellular and Molecular Medicine, Molecular Endocrinology Laboratory, KU Leuven, Leuven, Belgium. [99]Genomic Medicine Group, Galician Foundation of Genomic Medicine, Instituto de Investigacion Sanitaria de Santiago de Compostela (IDIS), Complejo Hospitalario Universitario de Santiago, Servicio Galego de Saúde, SERGAS, Santiago de Compostela, Spain. [100]University of California San Diego, Moores Cancer Center, La Jolla, CA, USA. [101]Division of Cancer Sciences, Manchester Cancer Research Centre, Faculty of Biology, Medicine and Health, Manchester Academic Health Science Centre, NIHR Manchester Biomedical Research Centre, Health Innovation Manchester, University of Manchester, Manchester, UK. [102]Case Western Reserve University, Department of Population and Quantitative Health Sciences, Cleveland Institute for Computational Biology, Cleveland, OH, USA. [103]Department of Clinical Chemistry, Erasmus University Medical Center, Rotterdam, The Netherlands. [104]Epidemiology and Biostatistics Unit, Centre Armand-Frappier Santé Biotechnologie, Institut National de la Recherche Scientifique, Laval, QC, Canada. [105]Department of Social and Preventive Medicine, School of Public Health, University of Montreal, Montreal, QC, Canada. [106]The University of Miami School of Medicine, Sylvester Comprehensive Cancer Center, Miami, FL, USA. [107]Nuffield Department of Surgical Sciences, University of Oxford, Oxford, UK. [108]NORMENT, KG Jebsen Centre, Oslo University Hospital and University of Oslo, Oslo, Norway. [109]Department of Radiology, University of California San Diego, La Jolla, CA, USA. [110]Department of Bioengineering, University of California San Diego, La Jolla, CA, USA. Lists of members and their affiliations appear in the Supplementary Information. ✉email: tseibert@ucsd.edu

