## [Peer Review File · Nature Communications]

Reviewers' Comments:

Reviewer #1:

Remarks to the Author:

The paper by Seibert et al looks to extend the polygenic risk score to other ancestries for their Prostate risk score.

Although the overall premise of the paper is sound, we find that the derivation of genetic ancestry groups is too crude and arbitrary. The current method used, which divides individuals into three broad groups, underestimates the complexity of human population structure. This is evidenced in eTable 2, where the relative richness of the self-reported ethnicity variable and the arbitrariness of the genetic sorting is clearly on display. While we encourage the use of genetically derived ancestry over self-reported, the current groups do not capture the finer grain structure or explore the complexity of how to handle human ancestry. The current methodology of grouping forces classification, even in circumstances where one of the three groups may not be applicable.

We appreciate the problem of wanting to have enough people in groups to know whether the risk scores work whilst also having some connection to genetic components, which ultimately is not a "group" concept but rather a complex tree (ultimately - ancestral recombination graph) makes this a fundamentally hard problem. Nevertheless this is the problem that the authors have decided to tackle.

Ideally one would explore different ways of handling this - for example a variety of granular views of groups or perhaps a continuous variable(s) to represent location (as advocated by some groups recently) or, at the very least, exploring different ways to make groups at different granularity levels.

We cannot recommend acceptance with this crude view of ancestry, but we do think a substantially revised paper, likely with similar outputs (though one never knows!) with some responsible exploration of the complexity of genetic ancestry is appropriate to be published.

Reviewer #2:

Remarks to the Author:

Generally, I found the manuscript clear and well written. I just have some minor comments.

Supplemental Table 2. The names of the SNPs is nonstandard. Please provide rs numbers where these are available and also give the chromosome and position for all markers. Chromosome_location is not unique because some positions have multiple variants at the same position. It is also helpful to give referent and alternate allele at each position. That will also help to identify any in/dels that may have been included.

Supplemental table 6-10, I think HR should be spelled out the first time it is used. Is it possible to combine these tables?

Where sample corrections were applied as described on line 428 it would be helpful to indicate for what populations (should require a short phrase).

What was the threshold of African, European and Asian ancestry used to define the continental origins? I did not see that indicated and it should be. What proportion of individuals could not be classified into a specific ethnic population?

Reviewer #1 (Remarks to the Author):

The paper by Seibert et al looks to extend the polygenic risk score to other ancestries for their Prostate risk score.

Thank you for the helpful reviewer comments. Please see below for our modifications to the manuscript.

Although the overall premise of the paper is sound, we find that the derivation of genetic ancestry groups is too crude and arbitrary. The current method used, which divides individuals into three broad groups, underestimates the complexity of human population structure. This is evidenced in eTable 2, where the relative richness of the self-reported ethnicity variable and the arbitrariness of the genetic sorting is clearly on display. While we encourage the use of genetically derived ancestry over self-reported, the current groups do not capture the finer grain structure or explore the complexity of how to handle human ancestry. The current methodology of grouping forces classification, even in circumstances where one of the three groups may not be applicable.

We appreciate the problem of wanting to have enough people in groups to know whether the risk scores work whilst also having some connection to genetic components, which ultimately is not a "group" concept but rather a complex tree (ultimately - ancestral recombination graph) makes this a fundamentally hard problem. Nevertheless this is the problem that the authors have decided to tackle.

Ideally one would explore different ways of handling this - for example a variety of granular views of groups or perhaps a continuous variable(s) to represent location (as advocated by some groups recently) or, at the very least, exploring different ways to make groups at different granularity levels.

We cannot recommend acceptance with this crude view of ancestry, but we do think a substantially revised paper, likely with similar outputs (though one never knows!) with some responsible exploration of the complexity of genetic ancestry is appropriate to be published.

We completely agree with the reviewer that human genetic ancestry is complex and multifaceted. In our initial submission, we used previously determined genetic ancestry from the **OncoArray** project¹, the source of the genotyping for the present work. However, we recognize that placing men into solely three genetic ancestry groups cannot be assumed to adequately account for the ancestral recombination graph, as the reviewer aptly describes it.

Consequently, we have now substantially expanded our subgroup analyses to further explore the diversity in our dataset and allow for multiple possibilities for genetic classification. The fastSTRUCTURE² algorithm was employed, an efficient schema which infers global admixture/ancestry via a Bayesian approach and generates a specified number (K) of agnostic clusters based on the genotypic data. We ran fastSTRUCTURE on the multi-ethnic dataset for multiple levels of population complexity to agnostically cluster the data into $K=2-6$ populations. PHS₂ was subsequently evaluated for association with aggressive prostate cancer (HR_{80/20}), stratified by all K population groupings. For these exploratory analyses, individuals were now assigned to a subpopulation if their maximum admixture proportion was ≥ 0.8 for any given subpopulation, and the remainder were assigned an otherwise unclassified "admixed" category.

Surprisingly, we found that the optimal fastSTRUCTURE model (i.e. the clustering that maximizes the marginal likelihood of the data) was the model with $K=2$ clusters. In that model, cluster 1 had mainly men of European OncoArray-derived genetic ancestry and self-reported race/ethnicity, cluster 2 had only men of African OncoArray-derived genetic ancestry and mostly Black/African American self-reported race/ancestry, while the Admixed cluster included men of all Oncotype-derived genetic ancestries. Consistent with the findings with OncoArray-defined genetic ancestry, the cluster with predominantly African ancestry (cluster 2) performed less well than cluster 1 (predominantly European) or the admixed cluster (**Table 6**).

Importantly, while not directly comparable, results were similar across models using different methods for evaluating ancestry and global admixture: the HR for aggressive PCa for the top vs. bottom quintile of PHS₂ was 2.1 in cluster 2, compared to 5.6 and 5.0 in cluster 1 and the admixed cluster, respectively. Results for analyses with $K=3-6$ are also reported in the **Supplemental Results**, providing a detailed look at various ways of subdividing the data by ancestry informative genotypic markers.

Finally, though self-reported race/ethnicity is not as biologically reliable, we also evaluated PHS performance using participants' self-reported race/ethnicity. Groups with $\geq 1,000$ men (to allow for adequate sample size³) included: European; Black or African (including African American, Black African, and Black Caribbean); and East Asian. These analyses are reported in the **Supplemental Results**.

Reviewer #2 (Remarks to the Author):

Generally, I found the manuscript clear and well written. I just have some minor comments.

Thank you for the complimentary remarks and the insightful comments below. Please see below for our revisions to the manuscript.

Supplemental Table 2. The names of the SNPs is nonstandard. Please provide rs numbers where these are available and also give the chromosome and position for all markers. Chromosome_location is not unique because some positions have multiple variants at the same position. It is also helpful to give referent and alternate allele at each position. That will also help to identify any in/dels that may have been included.

We have modified this supplemental table (now **eTable 5**) to show rs numbers for the SNPs. We also show the effect and reference alleles for each of these SNPs.

Supplemental table 6-10, I think HR should be spelled out the first time it is used. Is it possible to combine these tables?

We have now spelled out the abbreviation with first use in the supplemental tables. We have also condensed supplemental tables 6-10 as follows. The results of sensitivity analyses (previously supplemental tables 6-8) have been consolidated into new **supplemental table 7**. The results of the PHS and family history analyses (previously supplemental tables 9-10) have been consolidated into new **supplemental table 8**.

Where sample corrections were applied as described on line 428 it would be helpful to indicate for what populations (should require a short phrase).

We have clarified in the main manuscript (**Methods/Any prostate cancer**) that the sample corrections were derived from the Swedish population. Additional details for the sample weight corrections are provided in the **Supplemental Methods/Sample-Weight Correction and Sensitivity Analysis**.

What was the threshold of African, European and Asian ancestry used to define the continental origins? I did not see that indicated and it should be. What proportion of individuals could not be classified into a specific ethnic population?

The initial analysis used genetic ancestry categories defined previously by the OncoArray Consortium^{1,4} and provided with the PRACTICAL Consortium dataset. A description of this process was formerly in the Supplemental Materials, but we have now moved it to the main manuscript (**Methods/OncoArray-Defined Genetic Ancestry**). As now clearly stated, the thresholds chosen in that prior work depended on category to allow for a greater proportion of individuals to be classified: European: greater than 80% European ancestry, Asian: greater than 40% Asian ancestry, and African: greater than 20% African ancestry. In the dataset used here, all individuals were assigned one of the three continental classifications above.

However, in light of comments/suggestions by *Reviewer 1*, above, we have now substantially expanded our exploration of methods for evaluating ancestry and global admixture. We now include analyses of PHS performance in subcategories that include self-reported ethnicity/race and models of 2-6 agnostic genetic ancestry clusters, as well as admixed clusters. These results, along with those for self-reported race/ethnicity, are now reported in **Table 6** and the **Supplemental Material/Results**. Importantly, while not directly comparable, results were similar across models using different methods for evaluating ancestry and global admixture.

References

1. Amos CI, Dennis J, Wang Z, et al. The OncoArray consortium: A network for understanding the genetic architecture of common cancers. *Cancer Epidemiol Biomarkers Prev*. 2017;26(1):126-135. doi:10.1158/1055-9965.EPI-16-0106
2. Raj A, Stephens M, Pritchard JK. FastSTRUCTURE: Variational inference of population structure in large SNP data sets. *Genetics*. 2014;197(2):573-589. doi:10.1534/genetics.114.164350
3. Karunamuni RA, Huynh-Le M-P, Fan CC, et al. The effect of sample size on polygenic hazard models for prostate cancer. *Eur J Hum Genet*. June 2020. doi:10.1038/s41431-020-0664-2
4. Li Y, Byun J, Cai G, et al. FastPop: A rapid principal component derived method to infer intercontinental ancestry using genetic data. *BMC Bioinformatics*. 2016;17(1). doi:10.1186/s12859-016-0965-1

Reviewers' Comments:

Reviewer #1:

Remarks to the Author:

Apologies for the tardiness of my review.

As a reviewer of the first revision, the authors have answered my concerns as described - though I feel the authors could go further. I do think the authors are pulling apart some of the complexity in the ethnicity aspect in this paper. I note that although the $K=2$ might be optimal in the FastSTRUCTURE it looks to my reading of the $K=5$ and 6 in eTable 10 that the HR80/20 is better in these "high K " cases (indeed there is no poor HR80/20; I do find this a bit weird).

It might well be that the sample size is not good; that you are not happy exploring what this means because it opens more of the Pandora's box of what on earth is going on with ethnicity, and the fact that the clinical follow up is differentiated by ethnicity gets in the way.

My suggestion is to firmly acknowledge that a paper like this is just the start of understanding ethnicity wrt to Prostate GRS, and that one needs more details on both the genetics, but also on the "health seeking" and health delivery factors in society. I would also be on the side of the angels pointing out the accepted practices in medicine which has led to the asymmetrical data gathering (and likely different outcomes) which we must eliminate in society.

Strawman sentences from my perspective, consistent with the paper and this analysis (you might disagree, in which case argue with the editor how to represent table e10 appropriately):

"Although the optimal FastSTRUCTURE categories ($K=2$) produces a similar differentiation of PHS and HR80/20, we note that higher K structure numbers, although less representative of the data under the model produced a better range of HR80/20 hazards. The small sample sizes and complexity of reporting bias confounding means that these findings are not robust, but do suggest that deeper and more extensive studies with high granularity in groupings or more continuous models will be interesting to explore in the future. We also note the importance of consistent, deep clinical data collection across society to drive both research and better outcomes for all citizens in Prostate cancer research"

Minor points:

In both FastSTRUCTURE and self reported ethnicity, I would give a one sentence summary in the main paper rather than sending the reader to the supplement and back.

For eTable 10, can you put in the HR80/20 for $K=2$ as well so the comparison is easier to make?

I am content with the paper though I think there is still a mountain to climb to sorting all this out in the long term. This is not this paper's goal, but you should aim to acknowledge the task!

Signed:

Ewan Birney

Reviewer #2:

Remarks to the Author:

The authors have responded well to comments from the reviewers. I have a few minor suggestions.

1. I wonder if three significant digits can be used instead of just 2. The reason for this request is that the beta values that are given will be too imprecise for accurate subsequent computations should another reader want to apply them for meta-analysis or some other purpose.
2. Table e10, to make this table complete, I suggest also including 2 clusters even though that is duplicating a result presented in the main paper.

Definition of family history - this is not really given. Please clarify that you mean a family history of at least one first degree relative in a family or does family history vary among studies? If it does then you should spell out the definition for each study somewhere in the manuscript. Does family history also include family history in an identical twin? In that case it would be family history of prostate cancer in a first or lower degree relatives (but maybe this is not really relevant given the infrequency of identical twins).

REVIEWERS' COMMENTS

Reviewer #1, expert in bioinformatics/ethnicity, risk models (Remarks to the Author):

Apologies for the tardiness of my review.

As a reviewer of the first revision, the authors have answered my concerns as described - though I feel the authors could go further. I do think the authors are pulling apart some of the complexity in the ethnicity aspect in this paper. I note that although the $K=2$ might be optimal in the FastSTRUCTURE it looks to my reading of the $K=5$ and 6 in eTable 10 that the HR80/20 is better in these "high K " cases (indeed there is no poor HR80/20; I do find this a bit weird).

It might well be that the sample size is not good; that you are not happy exploring what this means because it opens more of the pandora's box of what on earth is going on with ethnicity, and the fact that the clinical follow up is differentiated by ethnicity gets in the way.

We agree that all of the HR80/20s derived from the fastSTRUCTURE agnostic ancestry groupings are generally quite good and demonstrate good risk-stratification for risk of aggressive prostate cancer. In that sense, we have achieved the primary aim of the study, which was to test the score derived in European data for performance in a large, independent, and more diverse dataset. However, we are in full agreement that this study does not attempt or offer a comprehensive understanding of the possible implications of race, ethnicity, or genetic ancestry. These are the subjects of ongoing further study by our group and others.

My suggestion is to firmly acknowledge that a paper like this is just the start of understanding ethnicity wrt to Prostate GRS, and that one needs more details on both the genetics, but also on the "health seeking" and health delivery factors in society. I would also be on the side of the angels pointing out the accepted practices in medicine which has led to the asymmetrical data gathering (and likely different outcomes) which we must eliminate in society.

Strawman sentences from my perspective, consistent with the paper and this analysis (you might disagree, in which case argue with the editor how to represent table e10 appropriately):

"Although the optimal FastSTRUCTURE categories ($K=2$) produces a similar differentiation of PHS and HR80/20, we note that higher K structure numbers, although less representative of the data under the model produced a better range of HR80/20 hazards. The small sample sizes and complexity of reporting bias confounding means that these findings are not robust, but do suggest that deeper and more extensive studies with high granularity in groupings or more continuous models will be interesting to explore in the future. We also note the importance of consistent, deep clinical data collection across society to drive both research and better outcomes for all citizens in Prostate cancer research"

We completely agree that ethnicity, race, and genetic ancestry are complex and multifaceted, particularly when attempting to correlate these factors with a patient's genetic risk of developing prostate cancer. Health disparities are also well established in prostate cancer epidemiology, treatment, and outcomes; addressing these issues is critical to improve overall care for men with

prostate cancer. There remains much more to learn regarding how these various factors influence polygenic prediction. We are especially interested in identifying subgroups where performance can be improved by ancestry-specific genetic markers. The suggested text, above, is appreciated. We have added these thoughts to the Discussion section.

Minor points:

In both FastSTRUCTURE and self reported ethnicity, I would give a one sentence summary in the main paper rather than sending the reader to the supplement and back.

This is an excellent suggestion. We have now put in a summary sentence to explain the cross tabulation of self-reported race/ethnicity and OncoArray-defined genetic ancestry in the main manuscript.

For eTable 10, can you put in the HR80/20 for $K=2$ as well so the comparison is easier to make?

We have added in the HR80/20's for $K=2$ in **Supplementary Table 10**.

I am content with the paper though I think there is still a mountain to climb to sorting all this out in the long term. This is not this paper's goal, but you should aim to acknowledge the task!

Thank you very much for your help to improve the quality of our paper! We are in agreement regarding the mountain ahead and are happy to acknowledge it in this first manuscript.

Signed:

Ewan Birney

Reviewer #2, expert in prostate cancer genetics and epidemiology (Remarks to the Author):

The authors have responded well to comments from the reviewers. I have a few minor suggestions.

Thank you very much for your helpful reviews and comments to improve our manuscript.

1. I wonder if three significant digits can be used instead of just 2. The reason for this request is that the beta values that are given will be too imprecise for accurate subsequent computations should another reader want to apply them for meta-analysis or some other purpose.

We have changed the results to include three significant digits.

2. Table e10, to make this table complete, I suggest also including 2 clusters even though that is duplicating a result presented in the main paper.

We have added in the HR80/20 for $K=2$ in **Supplementary Table 10**.

Definition of family history - this is not really given. Please clarify that you mean a family history of at least one first degree relative in a family or does family history vary among studies? If it does then you should spell out the definition for each study somewhere in the manuscript. Does family history also include family history in an identical twin? In that case it would be family history of prostate cancer in a first or lower degree relatives (but maybe this is not really relevant given the infrequency of identical twins).

Family history was standardized across studies included in the PRACTICAL consortium data. A positive family history of prostate cancer was defined as prostate cancer in at least one first degree relative. We have clarified in this definition of family history paper (**Methods**). The situation of an identical twin with prostate cancer is not specified in the PRACTICAL data dictionary.